# Learning to Confuse: Generating Training Time Adversarial Data with Auto-Encoder[*]

**Ji Feng**[1,2], **Qi-Zhi Cai**[2], **Zhi-Hua Zhou**[1]
[1]National Key Laboratory for Novel Software Technology
Nanjing University, Nanjing 210023, China
[2]Sinovation Ventures AI Institute
{fengj, zhouzh}@lamda.nju.edu.cn, caiqizhi@chuangxin.com

## Abstract

In this work, we consider one challenging training time attack by modifying training data with bounded perturbation, hoping to manipulate the behavior (both targeted or non-targeted) of any corresponding trained classifier during test time when facing clean samples. To achieve this, we proposed to use an auto-encoder-like network to generate such adversarial perturbations on the training data together with one imaginary victim differentiable classifier. The perturbation generator will learn to update its weights so as to produce the most harmful noise, aiming to cause the lowest performance for the victim classifier during test time. This can be formulated into a non-linear equality constrained optimization problem. Unlike GANs, solving such problem is computationally challenging, we then proposed a simple yet effective procedure to decouple the alternating updates for the two networks for stability. By teaching the perturbation generator to hijacking the training trajectory of the victim classifier, the generator can thus learn to move against the victim classifier step by step. The method proposed in this paper can be easily extended to the label specific setting where the attacker can manipulate the predictions of the victim classifier according to some predefined rules rather than only making wrong predictions. Experiments on various datasets including CIFAR-10 and a reduced version of ImageNet confirmed the effectiveness of the proposed method and empirical results showed that, such bounded perturbations have good transferability across different types of victim classifiers.

## 1 Introduction

How to modify the training data with bounded transferable perturbation that can lead to the largest generalization gap? In other words, we consider the task of adding imperceivable noises to the training data, hoping to maximally confuse any corresponding classifier so as to make wrong predictions as much as possible when facing clean test data. In this paper, we refer such perturbed training samples as training time adversarial training data.

To achieve the above goal, we defined a deep encoder-decoder-like network to generate such perturbations. Meanwhile, we used an imaginary neural network acting as the victim classifier, and the goal here is to train both networks simultaneously that can cause the lowest accuracy for the victim classifier on clean test set. We can thus formulate such problem into a non-linear equality constrained optimization problem. Unlike GANs [9], such optimization problem is much harder to solve, and a direct implementation of alternating updates will lead to unstable result. Inspired by some common techniques in reinforcement learning such as introducing a separate record tracking network like target-nets to stabilize Q-learning [19], we proposed a similar approach by decoupling the training

---

[*]The first two authors contributed equally to the work.

procedure for both networks. By doing so, the optimization procedure is much stable in practice. In other words, the adversarial perturbation generator is trained by hijacking the training procedure of the victim classifier. By doing so, the noise generator will learn to move against the victim classifier step by step.

A similar setting is data poisoning [20] proposed in the security community. However, their goal is quite different compared with this work. The main goal for this work is to reveal some intriguing properties of neural networks by adding bounded perturbations to the training data, whereas data poisoning focuses on the restriction that only a few training data is allowed to change. In other words, in traditional data poisoning tasks, the attackers goal is to add or modify training data as few as possible, whereas training time adversarial data put the constraint on the perturbation levels (as human imperceivable as possible). Moreover, having full control of training data (instead of changing a few) is a realistic assumption. For instance, in some applications an agent may agree to release some internal data for peer assessment or academic research, but does not like to enable the data receiver to build a model which performs well on real test data; this can be realized by applying such adversarial noises before the data release. In addition, when taking this from data privacy aspect, such procedure is quite different from releasing synthetic data via GANs. Consider a company selling surveillance cameras and the user will store all the data been taken (these photos cannot be synthetic for obvious reasons). On the other hand, the user certainly does not want any other unauthorized third parties to steal the data and train a classifier. Then, our proposed procedure is suitable for this kind of task since now the user can just make self-perturbations on its own data for protection.

The other contribution of this work is that, such formalization can be easily extended to the label specific case, where one wants to specifically fool the classifier of recognizing one input pattern into a *specifically predefined class*, rather than making a wrong prediction only. Finally, experimental results showed that, the learned noises is effective and robust to other machine learning models with different structure or even different types such as Random Forest [4] or Support Vector Machine(SVM) [6].

The rest of the paper is organized as follows: First, we will give a formalization for the proposed problem and describe the optimization procedure. Experimental results are then presented and finally conclusion and future works are discussed.

## 2   Related Works

One subject which closely relates to our work is data poisoning. The task of data poisoning dates back to the pre-deep learning times. For instance, there has been some research on poisoning the classical models, including SVM [2], Linear Regression [14], and Naive Bayes [21] which basically transform the poisoning task into a convex optimization problem.

Poisoning for deep models, however, is a more challenging one. Kon et.al. [16] first proposed the possibility of poisoning deep models via the influence function to derive adversarial training examples. Currently, there have been some popular approaches to data poisoning. For instance, sample specific poisoning aims to manipulate the model's behavior on some particular test samples. [24, 5, 11]. On the other hand, general poison attacks aiming to reduce the performance on cleaned whole unseen test set [16, 20]. As explained in the previous section, one of the differences with data poisoning is that the poisoning task mainly focuses on modifying as few samples as possible whereas our work focus on adding bounded noises as small as possible. In addition, our noise adding scheme can be scaled to much larger datasets with good transferability.

Another related subject is adversarial examples or testing time attacks, which refers to the case of presenting malicious testing samples to an already trained classifier. Since the classifier is given and fixed, there is no two-party game involved. Researches showed deep model is very sensitive to such adversarial examples due to the high-dimensionality of the input data and the linearity nature inside deep neural networks [10]. Some recent works showed such adversarial examples also exist in the physical world [8, 1], making it an important security and safety issue when designing high-stakes machine learning systems in an open and dynamic environment. Our work can be regarded as a training time analogy of adversarial examples. There have been some works on explaining the effectiveness of adversarial examples. The work in [26] proposed that it is the linearity inside neural networks that makes the decision boundary vulnerable in high dimensional space. Although beyond the scope of this paper, we tested several hypotheses on explaining the effectiveness of training time adversarial noises.

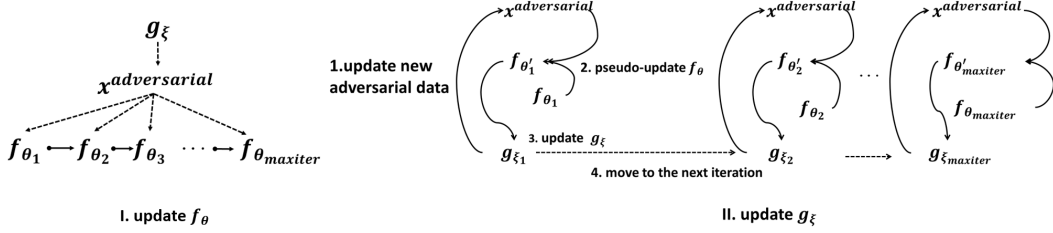

I. update $f_\theta$    II. update $g_\xi$

Figure 1: An overview for learning to confuse: Decoupling the alternating update for $f_\theta$ and $g_\xi$

## 3  The proposed method

Consider the standard supervised learning procedure for classification where one wants to learn the mapping $f_\theta : \mathcal{X} \to \{0,1\}^K$ from data where $K$ is the number of classes being predicted. To learn the optimal parameters $\theta^*$, a loss function such as cross-entropy for classification $\mathcal{L}(f_\theta(x), y) :$ $\mathbb{R}^k \times \mathbb{Z}_+ \to \mathbb{R}_+$ on training data is often defined and empirical risk minimization [27] can thus be applied, that is, one wants to minimize the loss function on training data as:

$$\theta^* = \arg\min_\theta \sum_{(x,y)\sim\mathcal{D}} [\mathcal{L}(f_\theta(x), y)] \tag{1}$$

When $f_\theta$ is a differentiable system such as neural networks, stochastic gradient descent (SGD) [3] or its variants can be applied by updating $\theta$ via gradient descent

$$\theta \leftarrow \theta - \alpha \nabla_\theta \mathcal{L}(f_\theta(x), y), \tag{2}$$

where $\alpha$ refers to the learning rate.

The goal for this work is to perturb the training data by adding artificially imperceivable noise such that during testing time, the classifier's behavior will be dramatically different on the clean test-set.

To formulate this, we first define a noise generator $g_\xi : \mathcal{X} \to \mathcal{X}$ which takes one training sample $x$ in $\mathcal{X}$ and transform it into an imperceivable noise pattern in the same space $\mathcal{X}$. For image data, such constraint can be formulated as:

$$\forall x, \|g_\xi(x)\|_\infty \leq \epsilon \tag{3}$$

Here, the $\epsilon$ controls the perturbation strength which is a common practice in adversarial settings [10]. In this work, we choose the noise generator $g_\xi$ to be an encoder-decoder neural network and the activation for the final layer is defined to be: $\epsilon \cdot (\tanh(\cdot))$ to facilitate the constraint (3).

With the above motivation and notations, we can then formalize the task into the following optimization problem as:

$$\begin{aligned} \max_\xi \quad & \sum_{(x,y)\sim\mathcal{D}} [\mathcal{L}(f_{\theta^*(\xi)}(x), y)], \\ s.t. \quad & \theta^*(\xi) = \arg\min_\theta \sum_{(x,y)\sim\mathcal{D}} [\mathcal{L}(f_\theta(x + g_\xi(x)), y)] \end{aligned} \tag{4}$$

In other words, every possible configuration $\xi$ is paired with one classifier $f_{\theta^*(\xi)}$ trained on the corresponding modified data, the goal here is to find a noise generator $g_{\xi^*}$ such that the paired classifier $f_{\theta^*(\xi^*)}$ to have the worst performance on the cleaned test set, compared with all the other possible $\xi$.

This non-convex optimization problem is challenging, especially due to the nonlinear equality constraint. Here we propose an alternating update procedure using some commonly accepted tricks in reinforcement learning for stability [19] which is simple yet effective in practice.

First, since we are assuming $f_\theta$ and $g_\xi$ to be neural networks, the equality constraint can be relaxed into

$$\theta_i = \theta_{i-1} - \alpha \cdot \nabla_{\theta_{i-1}} \mathcal{L}(f_{\theta_{i-1}}(x + g_\xi(x)), y) \tag{5}$$

where $i$ is the index for SGD updates.

Second, the basic idea is to alternatively update $f_\theta$ over adversarial training data via gradient descent and update $g_\xi$ over clean data via gradient ascent. The main problem is that, if we directly using this alternating approach, both networks $f_\theta$ and $g_\xi$ won't converge in practice. To stabilize this process, we propose to update $f_\theta$ over the adversarial training data first, while collecting the update trajectories for $f_\theta$, then, based on such trajectories, we update the adversarial training data as well as $g_\xi$ by calculating the pseudo-update for $f_\theta$ at each time step. Such whole procedure is repeated T trials until convergence. The detailed procedure is illustrated in Algorithm 1 and Figure 1.

---

**Algorithm 1:** Deep Confuse

**Input:** Training data $\mathcal{D}$, number of trials $T$, max iteration for training a classification model $maxiter$, learning rate of classification model $\alpha_f$, learning rate of the Noise Generator $\alpha_g$, batch size $b$

**Output:** Learned Noise Generator $g_\xi$

1   $\xi \leftarrow RandomInit()$
2   **for** $t = 1$ **to** $T$ **do**
3     $\theta_0 \leftarrow RandomInit()$
4     $L \leftarrow$ empty list
5     // Update $f_\theta$ while keeping $g_\xi$ fixed
6     **for** $i = 0$ **to** $maxiter$ **do**
7       $(x_i, y_i) \sim \mathcal{D}$ // Sample a mini-batch of training data
8       $L.append((\theta_i, x_i, y_i))$
9       $x_i^{adversarial} \leftarrow x_i + g_\xi(x_i)$
10      $\theta_{i+1} \leftarrow \theta_i - \alpha_f \nabla_{\theta_i} \mathcal{L}(f_{\theta_i}(x_i^{adversarial}), y_i)$ // Update model $f_\theta$ by SGD
11     **end**
12     // update $g_\xi$ via a pseudo-update of $f_\theta$
13     **for** $i = 0$ **to** $maxiter$ **do**
14       $(\theta_i, x_i, y_i) \leftarrow L[i]$
15       $\theta' \leftarrow \theta_i - \alpha_f \nabla_{\theta_i} \mathcal{L}(f_{\theta_i}(x_i + g_\xi(x_i)), y_i)$ // Pseudo-update $f_\theta$ over the current adversarial data
16       $\xi \leftarrow \xi + \alpha_g \nabla_\xi \mathcal{L}(f_{\theta'}(x), y)$ // Update $g_\xi$ over clean data
17     **end**
18   **end**
19   **return** $g_\xi$

---

Finally, we introduce one more modification for efficiency. Notice that storing the whole trajectory of the gradient updates when training $f_\theta$ is memory inefficient. To avoid directly storing such information, during each trial of training, we can create a copy of $g_\xi$ as $g'_\xi$ and let $g'_\xi$ to alternatively update with $f_\theta$, then copy the parameters back to $g_\xi$. By doing so, we can merge the two loops within each trial into a single one and don't need to store the gradients at all. The detailed procedure is illustrated in Algorithm 2.

---

**Algorithm 2:** Mem-Efficient Deep Confuse

**Input:** Training data $\mathcal{D}$, number of trials $T$, max iteration for training a classification model $maxiter$, learning rate of classification model $\alpha_f$, learning rate of the Noise Generator $\alpha_g$, batch size $b$

**Output:** Learned Noise Generator $g_\xi$

1   $\xi \leftarrow RandomInit()$
2   $g'_\xi \leftarrow g_\xi.copy()$
3   **for** $t = 1$ **to** $T$ **do**
4     $\theta_0 \leftarrow RandomInit()$
5     **for** $i = 0$ **to** $maxiter$ **do**
6       $(x_i, y_i) \sim \mathcal{D}$ // Sample a mini-batch
7       $\theta' \leftarrow \theta_i - \alpha_f \nabla_{\theta_i} \mathcal{L}(f_{\theta_i}(x_i + g'_\xi(x_i)), y_i)$ // Update $g'_\xi$ using current $f_\theta$
8       $\xi' \leftarrow \xi' + \alpha_g \nabla_{\xi'} \mathcal{L}(f_{\theta'}(x), y)$
9       $x_i^{adversarial} \leftarrow x_i + g_\xi(x_i)$
10      $\theta_{i+1} \leftarrow \theta_i - \alpha_f \nabla_{\theta_i} \mathcal{L}(f_{\theta_i}(x_i^{adversarial}), y_i)$ // Update $f_\theta$ by SGD
11     **end**
12     $g_\xi \leftarrow g'_\xi$
13   **end**
14   **return** $g_\xi$

---

## 4   Label Specific Adversaries

In this section, we give a brief introduction of how to transfer our settings to the label specific scenarios. The goal for label specific adversaries is that the adversary not only wants the classifier to make the wrong predictions but also want the classifier's predictions specifically according to some pre-defined rules. For instance, the attacker wants the classifier to wrongly recognize the pattern from class A specifically to Class B (thus not to Class C). To achieve this, denote $\eta : \mathbb{Z}_+ \to \mathbb{Z}_+$ as a pre-defined label transformation function which maps one label to another. Here $\eta$ is pre-defined by the attacker, and it transforms a label index into another different label index. Such label specific adversary can thus be formalized into:

$$
\begin{aligned}
\min_{\xi} \quad & \sum_{(x,y)\sim\mathcal{D}} [\mathcal{L}(f_{\theta^*(\xi)}(x), \eta(y))], \\
s.t. \quad & \theta^*(\xi) = \arg\min_{\theta} \sum_{(x,y)\sim\mathcal{D}} \mathcal{L}(f_\theta(x_i + g_\xi(x_i)), y_i)
\end{aligned}
\tag{6}
$$

It is easy to show that optimizing the above problem is nearly identical with the procedure described in Algorithm 2. The only thing needed to be changed is to replace the gradient ascent into gradient decent in line 10 in Algorithm 2 and replace $\eta(y)$ to $y$ in the same line while keeping others unchanged.

## 5   Experiment

To validate the effectiveness of our method, we used the classical MNIST [18], CIFAR-10 [17] for multi-classification and a subset of ImageNet [7] for 2-class classification. Concretely, we used a subset of ImageNet (bulbul v.s. jellyfish) consists of 2,600 colored images with size 224×224×3 for training and 100 colored images for testing. Random samples for the adversarial training data is illustrated in Figure 2.

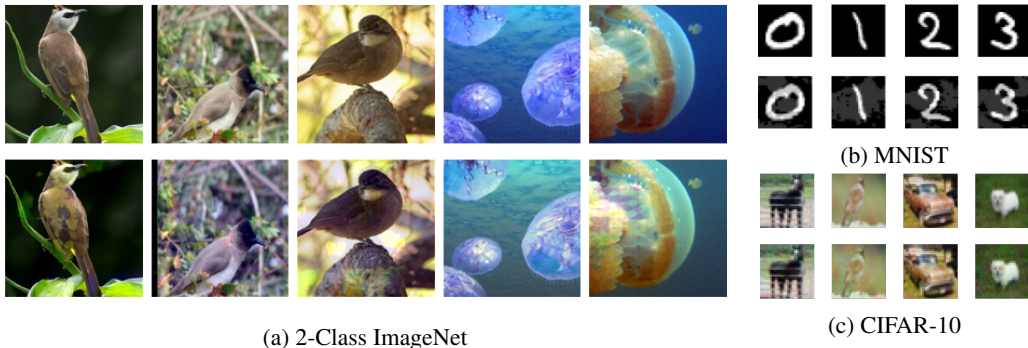

(a) 2-Class ImageNet

(b) MNIST

(c) CIFAR-10

Figure 2: First rows: original training samples. Second rows: adversarial training samples.

The classifier $f_\theta$ during training we used for MNIST is a simple Convolutional Network with 2 convolution layers having 20 and 50 channels respectively, followed by a fully-connected layer consists of 500 hidden units. For the 2-class ImageNet and CIFAR-10, we used $f_\theta$ to be a CNN with 5 convolution layers having 32,64,128,128 and 128 channels respectively, each convolution layer is followed by a 2×2 pooling operations. Both classifiers used ReLU as activation and the kernel size is set to be 3×3. Cross-entropy is used for loss function whereas the learning rate and batch size for the classifiers $f_\theta$ are set to be 0.01 and 64 for MNIST and CIFAR-10 and 0.1 and 32 for ImageNet. The number of trials $T$ is set to be 500 for both cases.

The noise generator $g_\xi$ for MNIST and ImageNet consists of an encoder-decoder structure where each encoder/decoder has 4 4x4 convolution layers with channel numbers 16,32,64,128 respectively. For CIFAR-10, we use a U-Net [23] which has larger model capacity. The learning rate for the noise generator $g_\xi$ is set to be $10^{-4}$ via Adam [15].

## 5.1 Performance Evaluation of Training Time Adversary

Using the model configurations described above, we trained the noise generator $g_\xi$ and its corresponding classifier $f_\theta$ with perturbation constraint $\epsilon$ to be 0.3, 0.1, 0.032, for MNIST, ImageNet and CIFAR-10, respectively. The classification results are summarized in Table 1. Each experiment is repeated 10 times.

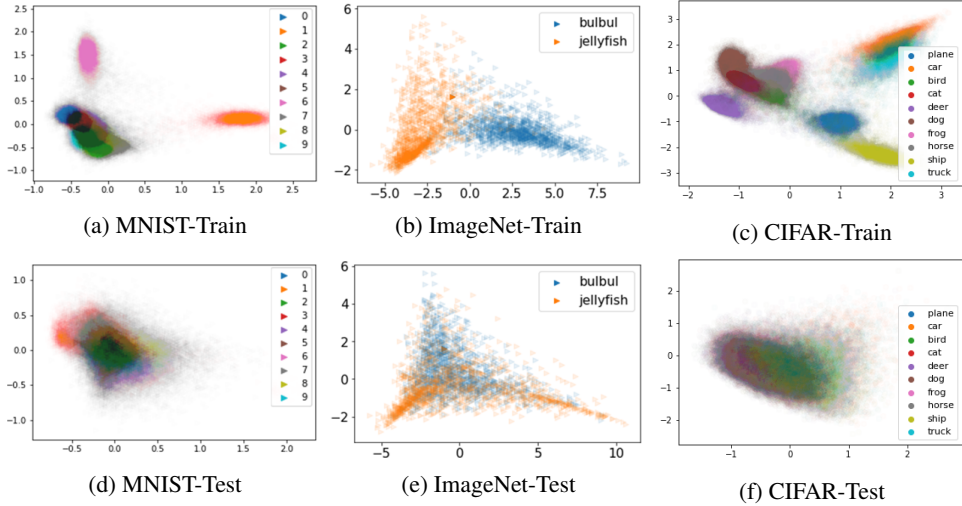

| (a) MNIST-Train | (b) ImageNet-Train | (c) CIFAR-Train |
|:---:|:---:|:---:|
| (d) MNIST-Test | (e) ImageNet-Test | (f) CIFAR-Test |

Figure 3: First row: Deep features of the adversarial training data. Second row: Deep features of the cleaned test data.

Table 1: Test accuracy (mean±std) when the classifier is trained on the original clean training set and the adversarial training set,respectively.

|  | MNIST | ImageNet | CIFAR-10 |
|---|---|---|---|
| Clean Data | $99.32 \pm 0.05$ | $88.5 \pm 2.32$ | $77.28 \pm 0.17$ |
| Adversarial Data | $0.25 \pm 0.04$ | $54.2 \pm 11.19$ | $28.77 \pm 2.80$ |

When trained on the adversarial datasets, the test accuracy dramatically dropped to only $0.25 \pm 0.04$, $54.2 \pm 11.19$ and $28.77 \pm 2.80$, a clear evidence of the effectiveness for the proposed method.

We also visualized the activation of the final hidden layers of $f_\theta$s trained on the adversarial training sets in Figure 3. Concretely, we fit a PCA [22] model on the final hidden layer's output for each $f_\theta$ on the adversarial training data, then using the same projection model, we projected the clean data into the same space. It can be shown that the classifier trained on the adversarial data cannot differentiate the clean samples.

It is interesting to know how does the perturbation constraint $\epsilon$ affects the performance in terms of both accuracy and visual appearance. Concretely, on MNIST dataset, we varied $\epsilon$ from 0 (no modification) to 0.3, with a step size of 0.05 while keeping other configurations the same and the results are illustrated in Figure 4.

The test accuracy in Figure 4 refers to the corresponding model performance trained on the different adversarial training data with different $\epsilon$. From the experimental result, we observed a sudden drop in performance when $\epsilon$ exceeds 0.15. Although beyond the scope of this work, we conjecture this result is related or somewhat consistent with a similar theoretical guarantee for the robust error bound when $\epsilon$ is 0.10 [28].

Finally, we examined the results when the training data is partially modified. Concretely, under different perturbation constraint, we varied the percentage of adversaries in the training data while keeping other configurations the same. The results are demonstrated in Figure 5. Random flip refers to the case when one randomly flip the labels in the training data.

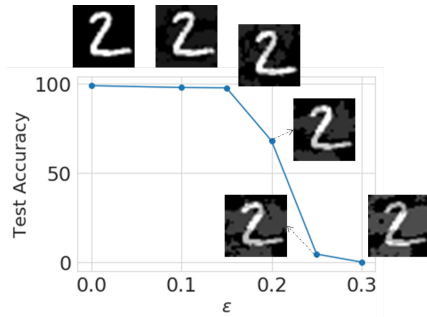

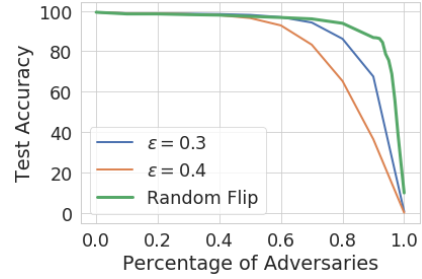

Figure 4: Effect of varying $\epsilon$.

Figure 5: Varying the ratio of adversaries under different $\epsilon$.

## 5.2 Evaluation of Transferability

In a more realistic setting, it is important to know the performance when we use a different classifier. Concretely, denote the original conv-net $f_\theta$ been used during training as $CNN_{original}$. After the adversarial data is obtained, we then train several different classifiers on the same adversarial data and evaluate their performance on the clean test set.

For MNIST, we doubled/halved all the channels/hidden units and denote the model as $CNN_{large}$ and $CNN_{small}$ accordingly. In addition, we also trained a standard Random Forest [4] with 300 trees and a SVM [6] using RBF kernels with kernel coefficient equal to 0.01. The experimental results are summarized in Figure 6.

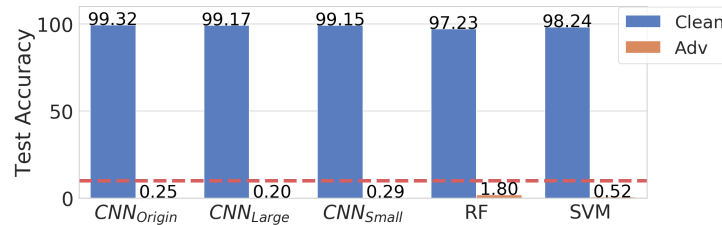

Figure 6: Test performance when using different classifiers. The horizontal red line indicates random guess accuracy.

The blue histograms in Figure 6 correspond to the test performance trained on the clean dataset, whereas orange histograms correspond to the test performance trained on the adversarial dataset. From the experimental results, it can be shown that the adversarial noises produced by $g_\xi$ are general enough such that even non-NN classifiers as random forest and SVM are also vulnerable and produce poor results as expected.

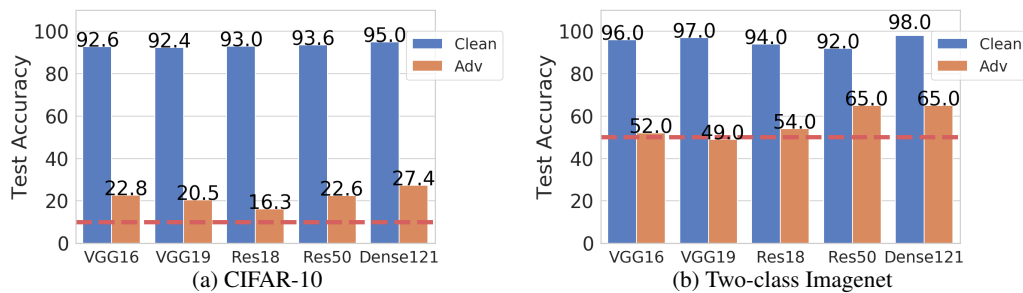

Figure 7: Test performance when using different model architectures. The horizontal red line indicates random guess accuracy.

For CIFAR-10 and ImageNet, we tried a variety of conv-nets including VGG [25], ResNet [12] and DenseNet [13] with different layers, and evaluate the performance accordingly. The results are summarized in Figure 7. Again, good transferability of the adversarial noise is observed.

## 5.3 The Generalization Gap and Linear Hypothesis

To fully illustrate the generalization gap caused by the adversaries, after we obtained the adversarial training data, we retrained 3 conv-nets (one for each data-sets) having the same architecture as $f_\theta$ and plotted the training curves as illustrated in Figure 8. A clear generalization gap between training and testing is observed. We conjecture the deep model tends to *over-fits towards the training noises* $g_\xi(x)$.

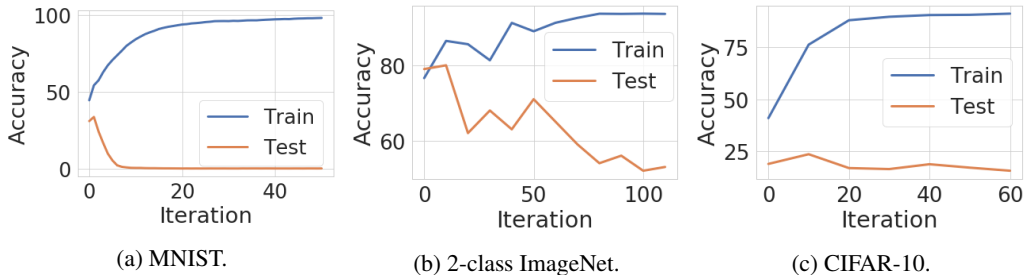

| (a) MNIST. | (b) 2-class ImageNet. | (c) CIFAR-10. |

Figure 8: Learning curves for $f_\theta$

To validate our conjecture, we measured the predictive accuracy between the true label and the *predictions $f_\theta(g_\xi(x))$ taking only adversarial noises as inputs*. The results are summarized in Table 2. Notice 95.15%, 93.00% and 72.98% test accuracy is obtained on the test set.

This interesting result confirmed the conjecture that the model does over-fit to the noises. Here we give one possible explanation. We hypothesize that it is the linearity inside deep models that make the adversarial effective. In other words, $f_\theta(g_\xi(x))$ contributes most when minimizing $\mathcal{L}(f_\theta(x + g_\xi(x)), y)$. This result is deeply related and consistent with the results from adversarial examples [10] and the memorization property for DNNs [29].

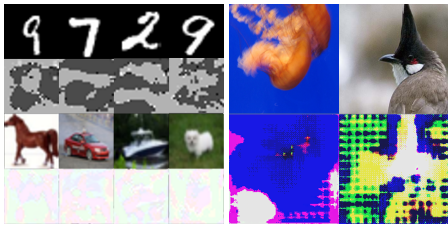

Figure 9: Clean samples and their corresponding adversarial noises for MNIST, CIFAR-10 and ImageNet

Table 2: Prediction accuracy taking **only noises as inputs**. That is, the accuracy between the true label and $f_\theta(g_\xi(x))$ where $x$ is the clean sample.

|  | Noise$_{\text{train}}$ | Noise$_{\text{test}}$ |
|---|---|---|
| MNIST | 95.62 | 95.15 |
| ImageNet | 88.87 | 93.00 |
| CIFAR-10 | 78.57 | 72.98 |

## 5.4 Weight Visualizations

Instead of visualizing deep features of the adversarial data, it is also interesting to directly plotting the trained weights of the victim classifier as a visual interpretation of the effectiveness. Concretely, we visualized the weights of two linear SVMs trained on clean and adversarial training data, respectively. Our results are shown in Figure 10.

It can be shown that, compared with image templates (top row) obtained from clean training data, the victim SVM weights (bottom row) trained on adversarial data went to the opposite direction and trend to over-fits on image corners. This result is also hinted that, the decision boundary in a high-dimensional space is indeed easy to manipulate, which in-turn give the attacker the chance of producing training time adversarial data.

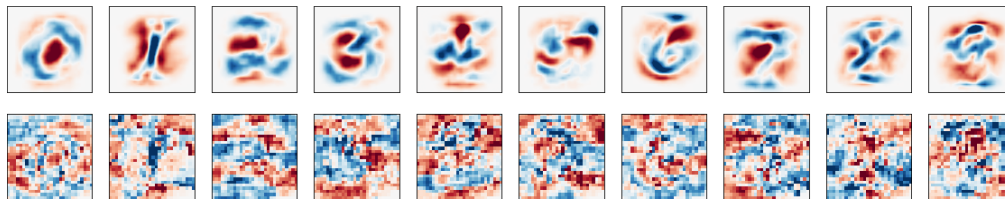

Figure 10: LinearSVM weights visualization for MNIST. Top row: Weights trained on clean training data. Bottom row: Weights trained on adversarial training data.

## 5.5 Label Specific Adversaries

To validate the effectiveness in label specific adversarial setting, without loss of generalizability, here we shift the predictions by one. For MNIST dataset, we want the classifier trained on the adversarial data to predict the test samples from class 1 *specifically* to class 2, and class 2 to class 3 ... and class 9 to class 0. Using the method described in section 4, we trained the corresponding noise generator and evaluated the corresponding CNN on the test set, as illustrated in Figure 11.

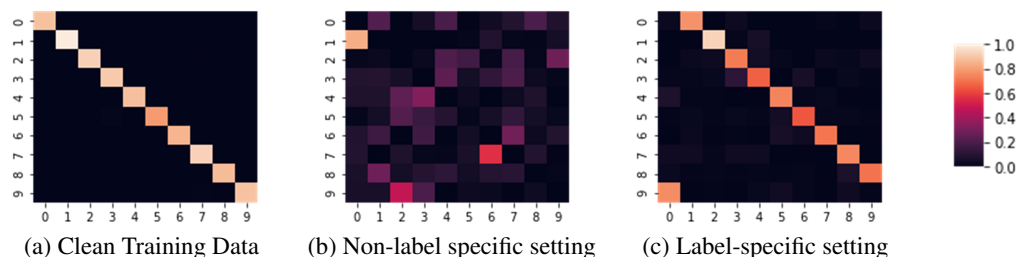

(a) Clean Training Data     (b) Non-label specific setting     (c) Label-specific setting

Figure 11: The confusion matrices on test set under different scenarios for MNIST dataset. They summarized the test performance of classifier trained on (a) clean training data (b) Non-label specific setting and (c) label-specific setting.

Compared with the test accuracy $(0.25 \pm 0.04)$ in the non-label specific setting, the test accuracy also dropped to $1.48 \pm 0.21$, in addition, the success rate for targeting the desired specific label increased from $0.00$ to $79.7 \pm 0.38$. Such results gave positive supports for the effectiveness in label specific adversarial setting. Notice this is only a side-product of the proposed method to show the formulation can be easily modified to achieve some more user-specific tasks.

## 6 Conclusion

In this work, we proposed a general framework for generating training time adversarial data by letting an auto-encoder watch and move against an imaginary victim classifier. We further proposed a simple yet effective training scheme to train both networks simultaneously by decoupling the alternating update procedure for stability. Experiments on image data confirmed the effectiveness of the proposed method, in particular, such adversarial data is still effective even to use a different victim classifier, making it more useful in a realistic setting.

Theoretical analysis or some more improvements for the optimization procedure is planned as future works. In addition, it is interesting to design adversarially robustness classifiers against this scheme.

### Acknowledgments

This research was supported by NSFC (61751306), the National Key R&D Program of China (2018YFB1004300), and the Collaborative Innovation Center of Novel Software Technology and Industrialization. The first two authors would like to thank Beijing Sinnovation Ventures Megvii International AI Institute Company Limited for the support.

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
