[Reviews · NeurIPS 2019]

Reviewer 1



1) The problem formulation is correct and it is backed well by supporting experimental results. But did not report any other baseline approaches and did not compare their framework to other approaches for this problem. 2) The writeup is good and easy to follow. Experiments are well described for others to repeat them. 3) The algorithm is novel and seems significant for others to continue from here.

Reviewer 2



[Edit after the author feedback]: I thank the authors for addressing my comments during the author feedback. I have read the authors' response as well as the other reviews. The authors' response addresses my concern regarding the motivation of the proposed method. Overall, I think this submission provides a new 'adversarial' setting, and I update my overall score as "6: Marginally above the acceptance threshold". ========================================================== Summary: This paper proposes a training time attack that is able to manipulate the behavior of trained classifiers, including deep models and classical models. The paper develops a heuristic algorithm to solve the expensive bilevel problem in Eq 4. The experimental results demonstrate the effectiveness of the proposed algorithm in this paper. Pros: - The proposed heuristic algorithm is efficient and able to find good solutions for the expensive bilevel problem. - The adversarial training examples generated by the proposed algorithm can manipulate the predictions of the classifiers and transfer between different types of classifiers. - The generated training time adversarial examples can transfer between different models including deep and non-deep models. Limitation & Questions: - Is there any comparable baseline method for this training time adversarial attack? - As shown in Figure 5, for the MNIST dataset, the model is 'robust' against the generated training adversarial examples when $\epsilon \leq 0.3$ and the percentage of adversaries is less than 60%. As 60% is already a large percentage, this makes the proposed attack less effective. Typo: - L230, should be $f_{\theta}(g_{\xi}(x))$. There is an omission in the related work on the bilevel problem in data poison attack: https://arxiv.org/abs/1811.00741

Reviewer 3



Post Response Comment: ========================================== I think the authors have addressed my initial concerns, therefore I maintain my initial stand and incline to accepting it. Originality ========================================= The setting is new as far as my knowledge can tell. Previous work such as "Certified Defense for Data Poisoning Attacks" considers contaminated instance within a feasible set, but modifying each training point by a small amount for an offline learner is new to me. I saw a backdoor attack in reference ([5]), but it is not referred to in the main body. I think the difference between this attack and the backdoor attack is that this one doesn't require the backdoor pattern to activate during test-time. Quality ========================================= The paper is technically sound overall. The most interesting part of the algorithm is using an encoder-decoder net instead of directly doing gradient ascent on the clean inputs to generate attack instances. Clarity ========================================= The writing can be more compact. There are running sentences here and there. Despite these flaws, the ideas are clearly conveyed. Significance ======================================= My main concern about the paper is the practicality of the setting. The author proposes a (hypothetical) setting in intro, but I'd like to see more explanation about the purpose. Is it to protect privacy? If so, how is it different from using synthetic data, say generated from GAN? I also have two suggestions. First, since SVM is also undermined by the attack, is it possible to visualize the SVM weights? Maybe it allows us to identify the pattern injected in the data. (Instead of a digit classifier, it might now detect whether a horizontal grey stroke is present.) Second, again for some convex model such as SVM or LR, the optimal model should satisfy some KKT condition (e.g. gradient=0), which can be a function of the training data. Maybe the conditions can shed light to how and why the training data are perturbed in some way.

[Author Response · NeurIPS 2019]

============ TO REVIEWER No.2 ============

[Q1.1] On similar approaches.

[A] Other than the discussions in the related works section, we wish to highlight that, the difference between our work and the usual "data poisoning" works is as follows: In data poisoning setting, the attacker's goal is to add or modify as few samples as possible (i.e., "an attacker usually cannot directly access an existing training database but may provide new training data" [2]), whereas our proposed framework put the constraint on the perturbation levels (as human imperceivable as possible). In addition, to our best knowledge, this is the first scalable solution that can run on big datasets (thanks to the optimization method we designed). Other SOTA methods for poisoning was only tested on datasets of small sizes (e.g., [20] used only 1,000 samples). In other words, a direct comparison is less fair for the standard poisoning algorithms. Nevertheless, we will include some more actual quantitative assessments for such comparison to make the points clearer in the revised version. Thank you for the suggestions. (Please also refer to the similar response to [Q2.1] and [Q3.2].)

============ TO REVIEWER No.4 ============

[Q2.1] On comparable baseline methods.

[A] Please also see [Q1.1]. We would also like to emphasize that, our work is not about poisoning only but rather on high-dimensional adversarial training examples with bounded perturbation. Therefore, we designed several different experiments with different angles toward this goal. Other existing methods for poisoning seems to be less aligned with our propose, making such comparisons less attractive. Meanwhile, as reviewer 2 also pointed out, we actually did conducted some baseline method such as Random Flip (Figure 5) for compare but only for validating purposes.

[Q2.2] "Is modifying 60% a practical setting?"

[A] Actually, yes. Unlike the traditional poisoning task, adversarial training data can be used in a good way. For instance, in some applications an agent may agree to release some internal data for academic research, but does not like to enable the data receiver to build a model which performs well on real test data; this can be realized by applying DeepConfuse before the data release. For motivations on privacy and why not just using synthetic data for these tasks, please also see our response to [Q3.2]. We will make the above points clearer in the revised version and to revise the typos/add the omitted related works as suggested. Thank you for the suggestions.

============ TO REVIEWER No.6 ============

[Q3.1] How about some visualizations and interpretations for the corresponding models?

[A] Thank you for the suggestion. For linear SVMs, we did a quick visualization on the weights trained on clean and adversarial training data as shown below:

Figure 1: LinearSVM weights visualization for MNIST. Top row: Weights trained on clean training data. Bottom Row: Weights trained on adversarial training data.

It can be shown that the weights for the SVM trained on adversarial data indeed went to the opposite direction compared with the corresponding clean model and trend to overfits on image corners. We will definitely add some more experimental results with a more detailed discussion in the revised version as suggested.

[Q3.2] On some more motivations.

[A] Please also see [Q1.1] and [Q2.1]. For data privacy aspect, our proposed approach is quite different from releasing synthetic data via GANs. Consider a company selling surveillance cameras and the user(usually the police) will store all the photos been taken. These photos cannot be synthetic for obvious reasons. On the other hand, such a company does not want any other third parties to train a classifier using their data. Then DeepConfuse is suitable for this kind of task and the camera company can just send the modified data in real-time. We will elaborate more on the motivation side in the revised version.

Thank you all for your thorough and insightful comments again.

[Meta-Review · NeurIPS 2019]

The paper proposes a novel algorithm to hijack the training process so the trained model performs very bad. This is an important topic and all the reviewers agreed that this paper should be accepted.